# Geographical variation and associated factors of vitamin A supplementation among 6–59-month children in Ethiopia

**Girma Gilano** [1] *, **Samuel Hailegebreal** [1], **Binyam Tariku Seboka** [2]

**1** Department of Health Informatics, School of Public Health, College of Medicine and Health Sciences, Arba Minch University, Arba Minch, Southern Ethiopia, **2** Department of Health Informatics, School of Public Health, College of Medicine and Health Sciences, Dilla University, Dilla, Southern Ethiopia

* gilanog@yahoo.com

## Abstract

### Introduction

Vitamin A has been one of the most important micronutrients which are necessary for the health of the children. In developing countries, the supplementation of vitamins under a regular schedule had different constraints. Awareness, access, and resource limitations were usually the problem. In the current study, we analyzed the data from the demographic health survey (EDHS) 2016 to uncover the spatial distribution, predictors, and to provide additional information for policymaking and interventions.

### Methods

In this analysis, we applied intra-community correlation to measure the random effect; global Moran's I to test the nature of variance in the null model; proportional change in variance to check the variance of null and neighborhood in subsequent models. We used STATA 15 for prediction; ArcGIS 10.7 for the spatial distribution of vitamin A supplementation; SaTscan 9.6.1 to specify location of clustering were the applied soft wares. After confirming that the traditional logistic regression cannot explore the variances, we applied multilevel logistic regression to examine predictors where p-value <0.25 was used to include variables into the model and p-value<0.05 was used to declare associations. We presented the result using means, standard deviations, numbers, and proportions or percent, and AOR with 95% CI.

### Result

The vitamin A coverage was 4,029.22 (44.90%) in Ethiopia in 2016. The distribution followed some spatial geo-locations where Afar, Somali were severely affected (RR = 1.46, P-value < 0.001), some pockets of Addis Ababa (RR = 1.47, p-value <0.001), and the poor distribution also affected all other regions partially. Place of delivery 1.2(1–1.34), primary and secondary education 1.3 (1–1.6), media exposure 1.2(1.1–1.4), having work 1.4(1.2–1.5), and all visits of ANC were positively influenced the distribution.

**Data Availability Statement:** The data used in this study are third-party data available from the Demographic and Health Survey (http://www.dhsprogram.com) and can be easily accessed by

following the protocol indicated in the methods and materials section.

**Funding:** The authors received no specific funding for this work.

**Competing interests:** I have read the journal's policy and the authors of this manuscript have the following competing interests: the authors declare that they have no competing interests.

## Conclusion

The distribution of vitamin A coverage was not random as per the EDHS 2016 data. Regions like Afar, Somali, and some pocket areas in Addis inquires immediate interventions. Pastoralist, agrarian, and city administrations were all involved from severe to the lesser coverage in order. Since factors like Place of delivery, education, ANC, media exposure, and having work were showed positive associations, interventions considering awareness, access, and availability of service need more attention than ever.

## Introduction

According to the World Health Organization (WHO), Vitamin A deficiency is a serious nutritional problem in developing and resources limited countries [1]. Among the impacts of vitamin A, Africa is the hugely affected continent with the highest blindness related to vitamin A deficiency [1]. In areas where vitamin A is the community problem, WHO recommends mass supplementation [2]. Vitamin A is such a crucial element in the application for malnutrition and also for prophylaxis. And it is readily available for use universally and also easy for solicitation [3]. Vitamin A supplementation decreases impaired vision, diarrhea, and mortality in children 6–59 months old [4]. Evidence also indicates that the severity of diarrhea reduces in children who used vitamin A previously. It also reduces admission and dehydration [5]. There has been variations between socioeconomically developed and underdeveloped countries with this regard [6]. Human beings obtain vitamin A from different sources of vegetables and fruits or sometimes grains [7, 8]. In Africa, vitamin A supplementation and immunization remained the problem which need solution until recently [9], and these problems worsen by the inequality of access to healthcare and socioeconomic factors which were an implication for child blindness [10]. Supplementation of vitamin A should be of an adequate doses as small doses showed lower effect [11]. In the African context, door-to-door supplementation is recommended for its success [12].

Malnutrition which includes vitamin A deprives healthy growth out of the children's life. In Ethiopia, provision of higher dose vitamin A showed association with high hemoglobin and low risk of anemia with family income makes the only differences among households [13, 14]. Vitamin A is the known public health problem in the countries where preschool children were not targeted for the supplementation [15]. Awareness creation is expected to work very well considering the socioeconomic factors of the communities. In Ethiopia, vitamin A distribution followed three strategies: enhanced outreach strategy, community health days, and the routine health extension program (regular community health workers) [16]. In the 2005 EDHS, the overall national coverage of vitamin A was 22.6%. It was 46.8% in preschool children in the same survey among 12–59 month children [17, 18]. It was also correlated with the socio-demographic factors of the child and the family [17]. According to the United Nation children Fund (UNICEF), every child has the right to survive and thrive- obtaining these limited services. The government of Ethiopia recognized and made malnutrition resulting from the likes of vitamin A, the national target to achieve millennium goals [19]. Despite all the various interventions in the country, vitamin A remained the major problem over decades [20]. Its effect remained the major health problems of the communities in Ethiopia which indeed requires identification of pockets and specific location with the currently influencing factors for further policy decisions and for the intensified implementation directions.

## Methods and materials

### Data source

This analysis utilized the data of the EDHS 2016. The approval letter to use the dataset was obtained from the Measure Demographic and Health Survey and the dataset was downloaded from the Measure DHS website: www.measuredhs.com.

### Study design and settings

We employed cross-sectional design data from EDHS 2016. The Ethiopia is divided into nine regions and 2 city administrations. It further categorized into three contextual regions as agrarian, pastoralists, and city administration. Tigray, Amhara, Oromia, Somali, Benishangul, Gambela, Harari, South Nation Nationalities People's Region (SNNPR) are agrarian regions. Afar and Somali are pastoralist regions; whereas Addis Ababa and Dire Dawa are city administrations [21]. EDHS 2016 dataset was requested for this study and the kids record (KR) was largely used for this study. EDHS collected data on fundamental traits of family health, including immunization coverage among children, prevalence and treatment of diarrhea and other diseases among children under five, and maternity care indicators such as antenatal visits and assistance at delivery. Biomarkers were collected for anemia from all children of age 6–59 months for whom consent was obtained from their parents or other adults responsible for them [22]. The total of 8,973 women with under-five children was used for the analysis.

EDHS used the frame of the population and housing census which contained the whole list of the evidence about the Enumeration Area (EA), location, type of residence (urban or rural), and the estimated number of residential households which were developed for this purpose by the Central Statistical Agency (CSA). A two-stage stratified cluster sampling technique was applied to amass the data. Initially, they established enumeration areas, which are the geographic locations that encompass enough number of dwelling elements that served as counting units in the census. In each enumeration area, they selected 28 households with the equal likelihood [22].

The supplementation of vitamin A in Ethiopia follows some strategic modalities like enhanced outreach strategy, community health days, and the routine health extension program. Enhanced outreach strategy was integrated in to the semi-annual or quarterly campaigns of deworming and malnutrition screening for implementation. Community health day is the campaign that has been organized at lower administrative (Kebele) levels. Community health day has been integrated in to the health extension program. Indeed currently, vitamin A supplementation is considered as the component of health extension program and majorly being disseminated by health extension (community health workers). The current study was not organized based on the models of supplementation. It was based on the nature of data available in DHS 2016 dataset [16].

### Study variables

The dependent variable is vitamin A supplementation. It was measured as present (Yes = 1) if the child was supplemented with vitamin A and not present (No = 0) if the child was not supplemented vitamin A in the last 6 months.

The independent variables includes individual level factors like the age of the mother, religion, marital status, the educational status of the mother, the educational status of the husband, working status, place of the delivery, sex of the child, the current age of the respondent, birth order, number of children live, and the number of antenatal care visits. And the community level factors like the region and the place of residence.

## Data management and analysis

The data were weighted considering sampling weight, primary sampling unit, and strata before descriptive analysis to restore the representativeness of the survey and to tell the STATA to take into account the sampling design when scheming standard errors, to get reliable statistical estimates.

## Spatial analysis

We used ArcGIS version 10.7 and Scan Statistics (SaTScanTM version 9.6.) softwares to execute the spatial data analysis. The presence of spatial autocorrelation was used to determine Global Moran's index (Moran's I). Hot-spot analysis was performed using Getis-Ord Gi* statistics. Spatial interpolation was also done to predict unmeasured areas of national vitamin A consumption based on the values from sampled data. Significant primary (most likely) and secondary clusters were identified by Spatial scan statistics. The outcome variable has Bernoulli distribution so the Bernoulli model was used by applying the Kuldorff method for purely spatial analysis. The default maximum spatial cluster size of $< 50\%$ of the population was used as an upper limit, which allowed both small and large clusters to be detected and ignored that contained more than the maximum limit. Areas with high Log-Likelihood Ratio (LR) and significant p-value were the areas with high vitamin A supplementation compared to areas outside the window.

## Multilevel analysis

Since the data showed a hierarchical nature, we assumed multilevel logistic regression. Four models were built from simplest (model 0/null) to more complex models step by step by checking the improvements at each level. Before adjusting for the variance through series of model development, we checked each variable at 0.25 p-values to include in the model. The final p-value remained $<0.05$ for the final model cut-point and AOR with 95% CI was also applied. Median Odds Ratio (MOR = $e^{0.95*} \sqrt{Va_{\_1}}$, where, $Va_{\_1}$ is the variance in the empty model) was examined [23]. The random effects were measured by Intra-class Correlation Coefficient (ICC) [24], and the Proportional Change in Variance (PCV) = $\frac{Va_{\_1} - Va_{\_2}}{Va_{\_1}}$, where, $Va_{\_1}$ is the variance of the empty model and $Va_{\_2}$ is neighborhood variance in the subsequent model) [23]. PCV was used to show the modification at the contextual level variance. We also used it between the empty model and the individual level model and between consecutive models. We applied deviance (-2LLR) to compare models. The lower the deviance the more fitted the model.

## Ethical approval and consent for participation

In this analysis, we used a secondary data that are publicly available from DHS. The EDHS protocol was reviewed and approved by the National Ethics Review Committee of the Federal Democratic Republic of Ethiopia, Ministry of Science, and Technology and the Institutional Review Board of ICF International. We obtained permission to access the data from the measure demographic and health survey through an online request form: http://www. measuredhsprogram.com. We got geographic coordinate data by elucidating the purpose of using the data, and then we received approval from the Measure DHS program. The DHS dataset is publically available; however, the term of use restricts any full or partial distribution of the dataset but figures that were obtained as an output of final analysis are not copy righted.

## Results

### Socio-demographic characteristics

We assembled the data from the weighted sample of 8,973 children aged 6–59 months who are appropriate for vitamin A supplementation as shown in Table 1. We found the number of respondents who reported vitamin A supplementation in the last six months 4,029.22 (44.90%). The mean age of the respondents was 29.5±6.5 months with 23.59% of children in

**Table 1. Socio-demographic characteristics of respondent among children aged 6–59 months in Ethiopia, EDHS 2016.**

| Variables | Weighted frequency (%) | Variables | Weighted frequency (%) |
|---|---|---|---|
| **Age in 5-yrs group** | | **Region** | |
| 15–19 | 233.5(2.60) | Tigray | 595.3 (6.63) |
| 20–24 | 1,630.75(18.17) | Afar | 93.11(1.04) |
| 25–29 | 2,763.2(30.79) | Amhara | 1,693(18.87) |
| 30–34 | 2,070.5(23.07) | Oromia | 3,886.8(43.31)) |
| 35–39 | 1,437.4(16.02) | Somali | 406.1(4.53) |
| 40–44 | 625.9(6.68) | Benishangul | 97.8(1.09) |
| 45–49 | 212.2(2.37) | SNNPR | 1,913.1(21.32) |
| | | Gambela | 22.14(0.25) |
| | | Harari | 20.9(0.23) |
| | | Addis Ababa | 205.8(2.29) |
| | | Dire Dawa | 39.08(0.44) |
| **Educational level** | | **Size of child at birth** | |
| No education | 5,956.3(66.38) | Small | 2,296.6(25.59) |
| Primary | 2,391.96(26.66) | Large | 2,926.33(32.61) |
| Secondary & Higher | 625.14(6.97) | Average | 3,750.5(41.80) |
| **Marital status** | | **Wealth status** | |
| Single | 45.96(0.51) | Poor | 4,204.6(46.86) |
| Married | 8,598.96(95.83) | Middle | 1,881.4(20.97) |
| Divorced | 106.54(1.19) | Rich | 2,887.4(32.18) |
| Divorced | 221.9(2.47) | | |
| **Media exposure** | | **Number of children** | |
| No | 6,043.01(67.34) | 1–2 | 7,357.09(81.99) |
| Yes | 2,930.38(32.66) | >3 | 1,616.3(18.01) |
| **Place of residence** | | **Current working status** | |
| urban | 1,002.02(11.17) | No | 6,432.717(71.69) |
| rural | 7,971.37(88.83) | Yes | 2,540.67(28.31) |
| **Sex of the child** | | ***VA in last 6 months** | |
| Male | 4,633.77(51.64) | No 46.25 56.75 | 4,944(55.10) |
| Female | 4,339.62(49.36) | Yes | 4,029.22(44.90) |
| **Religion** | | ***ANC visits** | |
| Orthodox | 3,119.87(34.77) | No visit | 5,213.11(58.10) |
| Muslim | 3,604.08(40.16) | 1st visit | 244.15(2.72) |
| protestant | 1,070.78(21.96) | 2nd visit | 445.83(4.97) |
| Others | 278.65(3.11) | 3rd visit | 1,132.33(12.62) |
| | | ≥4 visits | 1,937.97(21.60) |
| **Husband education** | | **Child age in months** | |
| No-education | 4,568.47(50.91) | 6–11 | 1,041.4(11.61) |

*(Continued)*

**Table 1.** (Continued)

| Variables | Weighted frequency (%) | Variables | Weighted frequency (%) |
|---|---|---|---|
| Primary education | 3,380.74(37.68) | 12–23 | 1,964.7(21.90) |
| Secondary & higher | 1,024.17(11.41) | 24–35 | 1,886.74(21.03) |
| | | 36–47 | 1,954.38(21.78) |
| | | 48–59 | 2,126.13(23.69) |
| **Place of delivery** | | | |
| Home | 6,792.05(74.69) | | |
| Institution | 2,271.34(25.31) | | |

*SNNPR = south nation nationalities peoples' region, VA = vitamin A, ANC = antenatal care.

the age group of 48–59 months. The largest (30.79%) number of respondents was in the age group 25–29 years. Male respondents were slightly higher (51%) than female. Most proportion participants were from Oromia (43.31%) followed by SNNPR (21.32%) and Amhara (18.87%). Over half (58.10%) of the respondents reported no ANC visit (ANC is the care that mother should obtain during pregnancy), and 46.86% lie in the poor economic category. Muslim is the largest religion (40.16%) among respondents, and 88.83% respondents were from rural communities. The majority of the respondents (67.34%) had no media exposure and had at least 1 or 2 children (81.99%). Most of them have no work (71.69%) during data collection time.

## Spatial analysis of vitamin A supplementation

**Spatial autocorrelation analysis.** This analysis showed that geospatial distribution of vitamin A supplementation was non-random in Ethiopia. The Global Moran's I test of 0.244 (P<0.05) observed in spatial autocorrelation analysis is an indication of the clustered pattern on the right side that showed high rate of vitamin A supplementation in the area (Fig 1).

**Hotspot analysis.** The local (Getis-Ord Gi*) statistics indicated that Tigray, Benshangul, Gambella, northern Afar, Eastern Oromia were identified as hotspot areas; whereas SNNP,

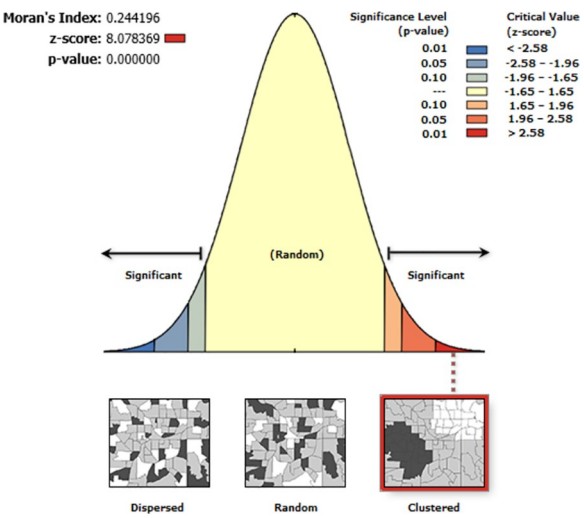

**Fig 1. Spatial autocorrelation of vitamin A supplementation among children aged 6–59 months in Ethiopia, EDHS 2016.**

central part of Afar, southern Oromia, Somali, and border of Amhara were identified as the cold spot regions of the country; (Fig 2).

**Spatial interpolation.** The ordinary kriging interpolation result revealed that regions such as Benishangul, most parts of Tigray, border of Amhara, the Eastern and Southern part of Gambela, western parts of SNNPR, western Oromia, and Northern Somali had higher rates of vitamin A supplement predicted. However, most parts of Somali, Oromia, and the Afar regions had lower predicted rate of vitamin A supplementation (Fig 3).

**Spatial scan statistical analysis.** One hundred fifty seven significant clusters (47 primary and 110 secondary locations) were identified in the SaTScan analysis. The primary spatial window was located in Somalia and Afar regions, which had the center at (7.717178 N, 46.991580 E) and geographic location with 555.85 km radius, and LLR of 85.4(p < 0.001). The relative risk (RR) of the primary clusters window was 1.46. This inferred as, the risk of not being supplied with vitamin A was 0.46 times higher among the children in the window (primary cluster) than those outside. There was also another cluster with coordinate/radius (11.555718 N, 41.433005 E) / 207.08 km located in central part of the country covering Addis Ababa largely and few parts of SNNP. The children in this window had 0.47 more risk of less vitamin A supplementation. There were other secondary clusters than this: two in Oromia, one in Amhara, two in Gambella, and one SNNP as displayed on Fig 4.

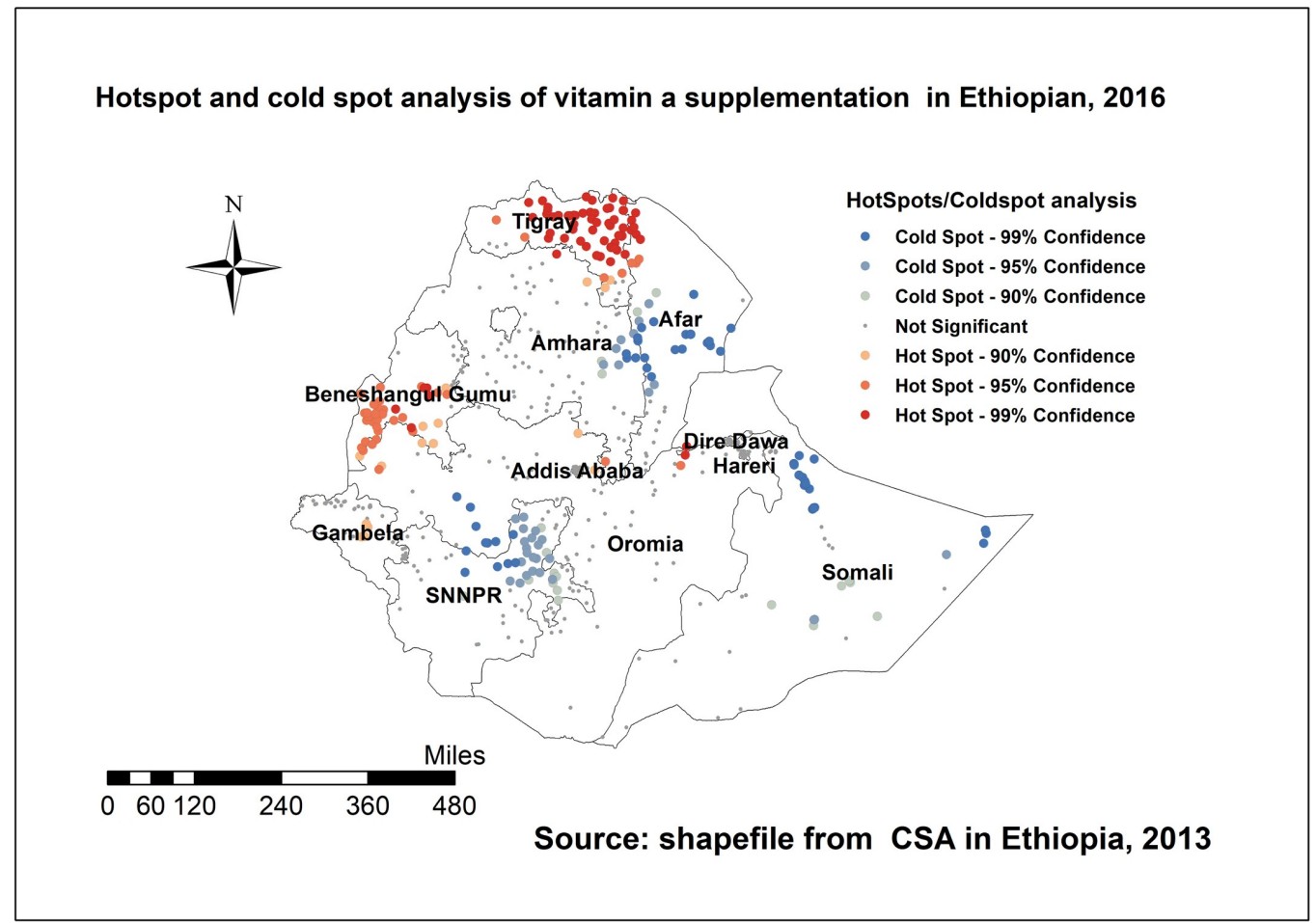

**Fig 2. Spatial distribution of vitamin A supplementation among children aged 6–59 months in Ethiopia, EDHS 2016.**

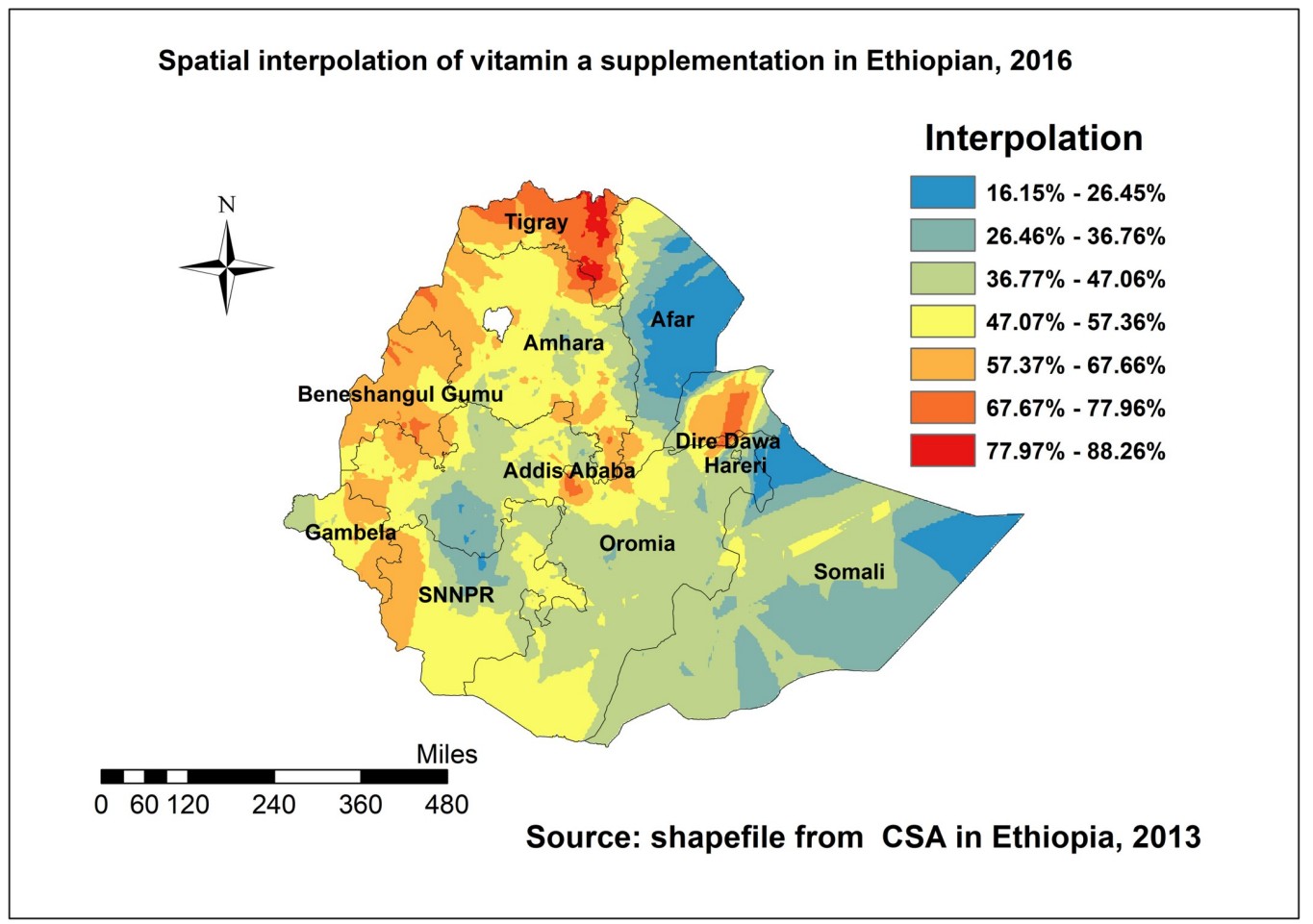

**Fig 3. Ordinary kriging interpolation of vitamin A supplementation among children aged 6–59 months in Ethiopia, EDHS 2016.**

**Random effect analysis.** Each of the four models built in this analysis improved performances. The community-level variance (ICC) was 26%, indicating that there was a significant difference in vitamin A supplementation at the community-level. The difference dwindled to 16% when we included and also controlled different variables; however, the unexplained community-level variance is remained higher in collective model which showed further investigation for other community-level factors might be necessary (Table 2).

During multilevel modeling: age, highest education level achieved, total number of children, preceding birth interval, literacy status, religion, wealth status, birth order, and region became significant both separately and in mixed effect model.

Factors like place of delivery, respondent's education, religion, media exposure, birth order, antenatal care, region and place of residence showed associations as shown in Table 3. In the mixed effect model, we added both individual and community-level predictors in to the multilevel logistic regression simultaneously. The odds of vitamin A supplementation were higher among those who gave birth at health institution compared to home delivery with AOR of 1.20 [1.0–1.34]. The odds of vitamin A supplementation was higher in respondents who learned primary and college or above education compared to no education with AOR of 1.20[1.02–1.34] and 1.30[1.00–1.60] respectively. Respondents who were in other religious category had higher odds of vitamin A supplementation compared to Orthodox Christians with AOR of

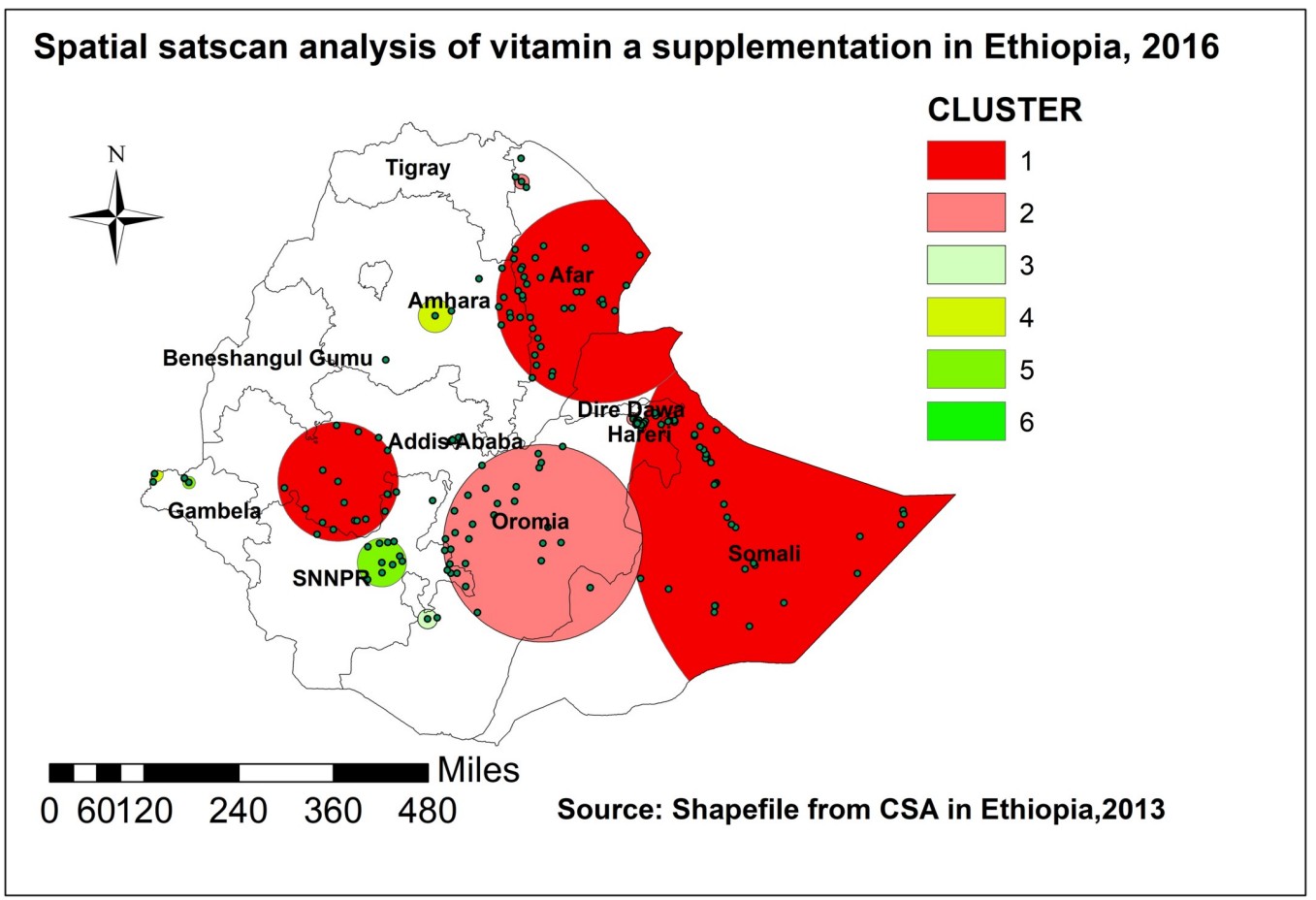

**Fig 4. SaTScan scan statistics of vitamin A supplementation among children aged 6–59 months in Ethiopia, EDHS 2016.**

1.60[1.10–2.34] and those with media exposure had similarly higher odds of vitamin A supplementation with AOR of 1.20[1.10–1.40] compared to non-media exposed. Women who had worked during data survey period were 1.40 times more likely to supplement vitamin A to their children with AOR of 1.40[1.20–1.50]. Odds of vitamin A supplementation among respondents who reported the presence of first, second, third and fourth antenatal visits was 1.40, 1.34, 1.56, and 1.60 times higher compared to those who reported no antenatal care with AOR of 1.40[1.10–1.90], 1.34[1.10–1.70], 1.56[1.3–1.80], and 1.60[1.40–1.80] respectively. Compared to Somali region, respondents from Tigray, Amhara, Benishangul, Gambela, and Dire Dawa regions had higher odds of vitamin A supplementation with AOR of 5[3.30–7.50], 1.60[1.10–2.00], 2.60[1.70–3.80], 1.70[1.10–2.50], and 3.60[1.30–5.60] respectively; (Table 3).

**Table 2. Model comparison and random effect distribution regarding vitamin A supplementation among children aged 6–59 months.**

| Random effect model comparison | Model 0 | Model 1 | Model 2 | Model 3 |
|---|---|---|---|---|
| Community-level variance | 1.18 | 0.93 | 0.68 | 0.63 |
| Inter-cluster correlation (ICC) | 0.26 | 0.22 | 0.17 | 0.16 |
| Log likelihood ratio (LLR) | -5544 | -5413 | -5429 | -5331 |
| Deviance | 11088 | 10826 | 10858 | 10662 |
| Proportional change in variance (PCV) | Ref | 0.21 | 0.42 | 0.46 |
| Media odds ratio (MOR) | 2.8 | 2.48 | 2.17 | 2.1 |

**Table 3. Multilevel logistic regression on vitamin A supplementation among children aged 6–59 months in Ethiopia, 2016.**

| Variables | Model 0 | Model I | Model II | Model III |
|---|---|---|---|---|
| **Place of delivery** | | | - | |
| Home | 1 | | - | |
| Health institution | 1.2(1.05–1.4) ** | | | 1.2(1–1.34) * |
| **Education of respondent** | - | | - | |
| No education | | 1 | | |
| primary | | 1.2(01.03–1.4) * | | 1.2(1.02–1.34) * |
| Secondary & above | | 1.2(0.97–1.5) | | 1.3(1–1.6) * |
| **Religion** | - | | - | |
| Orthodox | | 1 | | |
| Muslim | | 0.75(0.62–0.90) ** | | 1.1(0.87–1.34) |
| Protestant | | 0.94(0.76–1.12) | | 1.2(0.9–1.5) |
| Others | | 1.3(0.89–1.9) | | 1.6(1.1–2.34) * |
| **Media exposure** | - | | - | |
| No | | 1 | | |
| Yes | | 1.2(1-04-1.4) ** | | 1.2(1.1–1.4) ** |
| **Working status** | - | | - | |
| No | | 1 | | |
| Yes | | 1.4(1.22–1.6) | | 1.4(1.2–1.5) * |
| **Birth order** | - | 1.06(1–1.12) * | - | 1.04(1.0–1.1) |
| **Antenatal care visits** | - | | - | |
| No ANC | | 1 | | |
| 1st visit | | 1.4(1.06–1.9) * | | 1.4(1.1–1.9) * |
| 2nd visit | | 1.4(1.07–1.7) ** | | 1.34(1.1–1.7) * |
| 3rd visit | | 1.6(1.3–19) *** | | 1.56(1.3–1.8) *** |
| ≥4th visit | | 1.7(1.4–19) *** | | 1.6(1.4–1.8) *** |
| **Region** | - | - | | |
| Tigray | - | - | 7.4(5.1–10.7) *** | 5(3.3–7.5) *** |
| Afar | - | - | 0.7(0.52–1.09) | 0.7(0.49–1.03) |
| Amhara | - | - | 2.05(1.5–2.9) *** | 1.6(1.1–2.3) * |
| Oromia | - | - | 1.3(0.9–1.7) | 0.9(0.6–1.3) |
| Somali | - | - | 1 | - |
| Benishangul | - | - | 3.8(2.6–5.6) *** | 2.6(1.7–3.8) *** |
| SNNPR | - | - | 1.8(1.3–2.5) *** | 1.13(0.77–1.6) |
| Gambela | - | - | 2.5(1.7–3.7) *** | 1.7(1.1–2.5) * |
| Harari | - | - | 1.14(0.75–1.7) | 0.77(0.5–1.16) |
| Addis Ababa | - | - | 1.5(1–2.4) * | 0.9(0.6–1.5) |
| Dire Dawa | | | 4.7(3.1–7.4) | 3.6(1.3–5.6) *** |
| **Place of residence** | - | - | | |
| Rural | | | 1 | |
| Urban | | | 0.65(0.5–0.8) *** | 1.1(0.8–1.4) |

NB

\* = p<0.01

\*\* = p<0.05

\*\*\* = p<0.001.

## Discussion

In this study, among 8,973 pooled responses on vitamin A, only 44.90% of children aged 6–59 months had vitamin A supplementation in the last six months. There was an increase from previous studies prevalence that might be due to the make shift in strategies of delivery of vitamin A and integration of vitamin A delivery system in to the health extension program [2, 16]. There are more than a few factors associated with this at both individual and contextual levels. Media exposure, Religion, Education of the respondent, Place of delivery, and Antenatal care visits were amongst those predictors; however, age of the respondent and children, wealth index, birth order, and size of the child were not deemed significant here. According to the UNICEF, the magnitude of vitamin A supplementation in Ethiopia is half the coverage recommendations (≥80), but better than that of one study in Nigeria [10, 25]. The current coverage is also around half the country's national target of 80% [26]. The efforts made by responsible bodies for decades remained inadequate as per this finding. The respondents' wealth index was low. In statistical analyses, this was also accompanied by the significant association of lack of current work for the respondents which is connected with the low vitamin A supplementation. Evidence shows that wealth index and work status are the independent predictors of vitamin A supplementation; however, there is not enough evidence to support wealth index in our study [13, 16, 27].

Lack of information due to low educational status is usually connected to poor supplementation of vitamin A. In our study, 66.38% of respondents were not educated, while those who had primary education and above showed association with supplementing their children with vitamin A. It was not only mothers' education, but husband education had also a paramount value in reducing vitamin A supplementation. Here half of the husbands had no education, and the rest were also limited to primary level education; although, it was not evidenced in statistical analyses. There is enough evidence supporting the relationship between maternal education or family education and vitamin A supplementation [10, 27, 28]. Among the main finding of this study, children whose family or mother achieved some of the pre-delivery requirements like ANC follow-up and institutional delivery had a satisfied association with vitamin A more than others. It further confirmed that families or mothers who indeed engaged in service taking during pregnancy have a good tendency to endure service compelling after delivery. Pre-delivery services might increase the exposure to health professionals' advice. In contrast, they might be the only families or women that have access to services. This was also evidenced in the study conducted in the Dembia district(Amhara), where vitamin A deficiency was highly associated with the absence of ANC follow-up [15].

Spatial analysis portrayed that the pastoralists region were at higher risk of less vitamin A supplementation compared to other regions; however, poor vitamin A supplementation was also found severe in some pocket areas of the capital Addis. Previous studies were in line with lower vitamin A supplementation in pastoralists regions but in conflict with the finding of lower vitamin A supplementation in Addis Ababa [18, 27, 28]. The pocket areas uncovered by this analysis might be existed for a longtime unrecognized and now must draw attention for full coverage. The Afar and Somali (pastoralists) regions might be always at risk because of the poor resource distribution and less access to the services depending on the way of life. The finding of spatial distribution is well in line with the results displayed in Table 3 (statistical analysis). Vitamin A supplementation analysis also showed poor distribution in the Oromia region which is consistent with cumulative decreasing regional micronutrients coverage report from 2007 to 2011 and other reasons [17, 29]. Few parts of SNNP and Amhara also showed some vitamin A supplementation problems. There are different studies in the country that

showed these regions also suffering in some of their woredas from vitamin A supplementation-related problems [15, 17, 20].

More vitamin A supplementation problems were found in many regions, which still infer a deep-rooted issue in the country. Although this study is so crucial to portrait overall country-level information regarding vitamin A supplementation, it has also some limitations that might need consideration when using it. The disproportionate sampling, secondary nature of the data, third party data, and the need of referring some variable back to 2016 are some. To handle these problems, the authors applied weighting to the data, involved multilevel analysis, used only the data with full information, and secured full permissions.

## Conclusion

Overall the vitamin A supplementation in Ethiopia has remained below par and the distribution of risk followed the contextual nature of the regions. The highest risk was observed in Pastoralists regions, but agrarian regions were also involved vastly. Afar and Somali were the regions which highly affected followed by the Oromia region, some parts of SNNP, Amhara regions, and even some pocket areas of Addis Ababa. This spatial distribution might be very fundamental for policy makers and next interventions. Resource distribution might need to consult this information for optimal achievements. In other words, both individual and community factors had their hands in poor supplementation or service uptake in each region. The fact that the spatial distribution and statistical association were well supported one another showed the distribution is scientifically sound. And since the factors like Place of delivery, education, ANC, media exposure, and having work have positively prejudiced the vitamin A supplementation. Access, availability, application of vitamin A delivery system consistently and awareness might be the crucial tools to tackle the problem.

## Acknowledgments

The authors are very thankful for the responsible parties of the EDHS for the permission to download the data and all other stakeholders.

## Author Contributions

**Conceptualization:** Girma Gilano, Samuel Hailegebreal.

**Formal analysis:** Girma Gilano, Binyam Tariku Seboka.

**Methodology:** Girma Gilano, Samuel Hailegebreal, Binyam Tariku Seboka.

**Project administration:** Girma Gilano.

**Software:** Samuel Hailegebreal.

**Writing – original draft:** Girma Gilano, Binyam Tariku Seboka.

**Writing – review & editing:** Samuel Hailegebreal, Binyam Tariku Seboka.

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
