## [Decision Letter · Decision Letter 0]

28 Jul 2021

PONE-D-21-15858

Geographical variation and associated factors of vitamin A supplementation among 6–59-month children in Ethiopia

PLOS ONE

Dear Dr. Gilano,

Thank you for submitting your manuscript to PLOS ONE. After careful consideration, we feel that it has merit but does not fully meet PLOS ONE’s publication criteria as it currently stands. Therefore, we invite you to submit a revised version of the manuscript that addresses the points raised during the review process.

The study is interesting but would benefit will more in depth description of the delivery model that is used for vitamin A supplementation. Because this delivery model is closely linked with coverage and needs a rationale why coverage is high  related with the selected determinants. Secondly English editing is a must before resubmission as well as formatting references.

We look forward to receiving your revised manuscript.

Kind regards,

Mahfuzar Rahman, MD, PhD

Academic Editor

PLOS ONE

“We received no funds for this work.”

4. We note that Figures 2,3 & 4 in your submission contain [map/satellite] images which may be copyrighted. All PLOS content is published under the Creative Commons Attribution License (CC BY 4.0), which means that the manuscript, images, and Supporting Information files will be freely available online, and any third party is permitted to access, download, copy, distribute, and use these materials in any way, even commercially, with proper attribution. For these reasons, we cannot publish previously copyrighted maps or satellite images created using proprietary data, such as Google software (Google Maps, Street View, and Earth). For more information, see our copyright guidelines: http://journals.plos.org/plosone/s/licenses-and-copyright.

a) You may seek permission from the original copyright holder of Figures 2,3 & 4 to publish the content specifically under the CC BY 4.0 license. 

Additional Editor Comments (if provided):

Reviewers' comments:

Reviewer's Responses to Questions

**Comments to the Author**

1. Is the manuscript technically sound, and do the data support the conclusions?

Reviewer #1: Yes

Reviewer #2: Yes

2. Has the statistical analysis been performed appropriately and rigorously? 

Reviewer #1: Yes

Reviewer #2: Yes

3. Have the authors made all data underlying the findings in their manuscript fully available?

Reviewer #1: No

Reviewer #2: Yes

4. Is the manuscript presented in an intelligible fashion and written in standard English?

Reviewer #1: No

Reviewer #2: No

5. Review Comments to the Author

Reviewer #1: PLOSone 2021: Gilano et al.

General: This paper used Demographic Health Survey data from Ethiopia to determine the distribution of vitamin A supplementation. The use of secondary analysis is useful. The paper is poorly edited and the references are a mess. More care needs to be taken with the revision of the article.

L. 21: Capitalize “Vitamin” and use “that is” instead of “which are”

L. 36: delete “was”

L. 39: delete “have”

L. 41-42: The sentence is poorly constructed. Perhaps delete “was poor”.

L. 51: World Health Organization (WHO), vitamin A …

L. 54: Delete world health organization

Throughout: Do not capitalize vitamin A unless it is starting the sentence, i.e., L. 60, 154, 283

L. 68: Not sure what this phrase means.

L. 74: United Nations Children Fund

L. 78-9: Delete the second sentence starting with “Despite all” because it is redundant.

L. 86: Delete “Ethiopian Demographic Health survey” because it is already defined.

L. 93: “collected”

L. 144: “is” should be “are”

L. 159: Define “ANC” here.

L. 163: “.” At the end of the sentence.

l. 170: Ethiopia.

L. 174: “northern”

L. 175: “areas”

L. 176: “cold spot regions”

L. 180 and 181 and 197: Capital “A”

Table 2: No need to capitalize “variance”.

L. 204: Delete “were”

L. 213-4: “Women who had worked during the data survey period were 1.4 times more likely to…”

L. 237: “no” should be “not”

L. 239: Instead of “not learned” use “were uneducated”

L. 261: Delete “was”

L. 266-7: “More vitamin A supplementation problems were found in many regions, which still infers a deep-rooted issue in Ethiopia.”

L. 275: Delete the first “was”

The reference style is inconsistent.

Reference 3 should be “Ross DA”.

Some titles have all capital letters and others do not.

Reference 5 and 7 and 17 and 19 and 20 do not have a journal.

Reference 14 needs a capital “World Bank”.

Reference 18 has complete names and not initials. Reference 27 is the same as 18.

Figures 2 and 3 and 4: “A” needs to be capitalized on the figure.

Reviewer #2: the study is interesting but would benefit from reviewing the english with an english first language writer, and also will need more in depth description of the delivery model used for vitamin A supplementation, as it may be closely linked with coverage and why coverage is related to the selected determinants.

6. PLOS authors have the option to publish the peer review history of their article (what does this mean?). If published, this will include your full peer review and any attached files.

Reviewer #1: No

Reviewer #2: No

---

## [Author Response · Author response to Decision Letter 0]

6 Aug 2021

We are very thankful for the journal, editorial office and reviewers who gave their time to improve this work. We tried to admit every point rose by the editorial office and reviewers unless there might be a misunderstanding of the points. The detailed responses are as follow. 

Reviewer #1

Comment: L. 21: Capitalize “Vitamin” and use “that is” instead of “which are 

Response: done…. Vitamin A has been one of the most important micronutrients that is necessary for the health of children

Reviewer #1

Comment: L. 36: delete “was”

Response: The distribution followed some spatial geo-locations Reviewer #1

Comment: L. 39: delete “have”

Response: Place of delivery 1.2(1-1.34), secondary and education 1.3 (1-1.6), media exposure 1.2(1.1-1.4), having work 1.4(1.2-1.5), and all visits of ANC positively influenced the distribution.

Reviewer #1

Comment: 41-42: The sentence is poorly constructed. Perhaps delete “was poor”.

Response: The distribution of vitamin A supplementation coverage was not random according to the data from the Ethiopian demographic health survey 

Reviewer #1

Comment: L. 51: World Health Organization (WHO), vitamin A …

Response: World Health Organization (WHO)

Reviewer #1

Comment: L. 54: Delete world health organization

Response: In areas where vitamin A is a community problem, WHO recommends mass supplementation

Reviewer #1

Comment: Throughout: Do not capitalize vitamin A unless it is starting the sentence, i.e., L. 60, 154, 283

Response: unnecessarily capitalized vitamin A corrected

Reviewer #1

Comment: L. 68: Not sure what this phrase means.

Response: 

Reviewer #1

Comment: L. 74: United Nations Children Fund

Response: Vitamin A is the known public health problem in the countries where preschool children were not targeted for the supplementation

Reviewer #1

Comment: L. 78-9: Delete the second sentence starting with “Despite all” because it is redundant.

Response: Its effect remained the major health problems of the communities in Ethiopia which indeed requires identification of pockets for further policy for intensified implementation directions.

Reviewer #1

Comment: L. 86: Delete “Ethiopian Demographic Health survey” because it is already defined.

Response: The approval letter to use the dataset was obtained from the DHS and the dataset

Reviewer #1

Comment: L. 93: “collected”

Response: EDHS Collected data on fundamental traits of family health, including immunization…

Reviewer #1

Comment: L. 144: “is” should be “are”

Response: In this analysis, we used secondary data that are publicly available from EDHS 

Reviewer #1

Comment: L. 159: Define “ANC” here.

Response: Over half (58.10%) of the respondents reported no ANC visit (ANC is the care that mother should obtain during pregnancy),

Reviewer #1

Comment: L. 163: “.” At the end of the sentence.

Response: They were also not working (71.69%) during data collection time.

Reviewer #1

Comment: 170: Ethiopia.

Response: This analysis showed that geospatial distribution of vitamin A supplementation was non-random in Ethiopia

Reviewer #1

Comment: L. 174:“northern”

Response: The local (Getis-Ord Gi*) statistics indicated that Tigray, Benshangul, Gambella, northern Afar, Eastern Oromia were identified as hotspot area; whereas SNNP, central part of Afar, southern Oromia, Somali, and border of Amhara were identified as cold spots region of the country

Reviewer #1

Comment: L. 175: “areas”

Response: The local (Getis-Ord Gi*) statistics indicated that Tigray, Benshangul, Gambella, norther Afar, Eastern Oromia were identified as hotspot areas; whereas SNNP, central part of Afar, southern Oromia, Somali, and border of Amhara were identified as cold spots region of the country

Reviewer #1

Comment: L. 176: “cold spot regions”

Response: The local (Getis-Ord Gi*) statistics indicated that Tigray, Benshangul, Gambella, norther Afar, Eastern Oromia were identified as hotspot area; whereas SNNP, central part of Afar, southern Oromia, Somali, and border of Amhara were identified as cold spots regions of the country

Reviewer #1

Comment: L. 180 and 181 and 197: Capital “A”

Response: vitamin a was changed vitamin A

Reviewer #1

Comment: Table 2: No need to capitalize “variance”.

Response: Community-level variance

Reviewer #1

Comment: L. 204: Delete “were”

Response: ……literacy status, religion, wealth status, birth order, and region became significant both separately and in mixed effect model.

Reviewer #1

Comment: L. 213-4: “Women who had worked during the data survey period were 1.4 times more likely to…”

Response: Women who had worked during data survey period were 1.40 times more likely to supplement vitamin A to their children with AOR of 1.40[1.20-1.50].

Reviewer #1

Comment: L. 237: “no” should be “not”

Response: The respondents’ wealth index was low. In statistical analyses, this was also accompanied by the significant association of lack of current work for the respondent which is connected with low vitamin A supplementation. Evidence shows that wealth index and work status are the independent predictors of vitamin A supplementation; however, there is not enough evidence to support wealth index in our study.

Reviewer #1

Comment: L. 239: Instead of “not learned” use “were uneducated”

Response: In our study, 66.38% of respondents were not educated, while those who learned…

Reviewer #1

Comment: L. 261: Delete “was”

Response: Vitamin A supplementation analysis also showed poor distribution in the Oromia region which is consistent with cumulative decreasing regional micronutrients coverage report from 2007 to 2011

Reviewer #1

Comment: L. 266-7: “More vitamin A supplementation problems were found in many regions, which still infers a deep-rooted issue in Ethiopia.”

Response: Accepted

Comment: L. 275: Delete the first “was”

Response: Overall the vitamin A supplementation in Ethiopia has remained below par and the distribution of risk followed the contextual nature of the regions

Reviewer #1

Comment: Reference 3 should be “Ross DA”.

Response: Ross DA., Recommendations for Vitamin A Supplementation, The Journal of Nutrition, Volume 132, Issue 9, September 2002, Pages 2902S–2906S,

Reviewer #1

Comment: Some titles have all capital letters and others do not.

Response: all references were checked and corrected for spelling and capitalization

Reviewer #1

Comment: Reference 5 and 7 and 17 and 19 and 20 do not have a journal.

Response: name of the journal were added to the references

Reviewer #1

Comment: Reference 14 needs a capital “World Bank”.

Response: World Bank Group. Ethiopia Nutrition project performance Asessment Report. 2019;(136172):9. Available from:

Reviewer #1

 Comment: Reference 18 has complete names and not initials. Reference 27 is the same as 18

Response: Semba RD, Pee SD, Sun K, Bloem MW, Raju VK, Coverage of the National Vitamin A Supplementation Program in Ethiopia, Journal of Tropical Pediatrics, Volume 54, Issue 2, April 2008, Pages 141–144, https://doi.org/10.1093/tropej/fm.

Reviewer #1

Comment: Figures 2 and 3 and 4: “A” needs to be capitalized on the figure.

Response: 

Reviewer #2

Comment: reviewing the English 

Response: the whole document was reviewed for English with local professional editor

Reviewer #2

Comment: in depth description of the delivery model used for vitamin A supplementation 

Response: In Ethiopia, vitamin A distribution followed three strategies: enhanced outreach strategy, community health days, and the routine health extension program (regular community health workers)

The supplementation of vitamin A in Ethiopia follows some strategic modalities like enhanced outreach strategy, community health days, and the routine health extension program. Enhanced outreach strategy was integrated in to the semi-annual or quarterly campaigns of deworming and malnutrition screening for implementation. Community health day is the campaign that has been organized at lower administrative (Kebele) levels. Community health day has been integrated in to the health extension program. Indeed currently, vitamin A supplementation is considered as the component of health extension program and majorly being disseminated by health extension (community health workers). 

The increase from previous studies prevalence might be due the make shift in strategies of delivery of vitamin A and integration of vitamin A delivery system in to health extension program

---

## [Editor Report · Decision Letter 1]

15 Dec 2021

Geographical variation and associated factors of vitamin A supplementation among 6–59-month children in Ethiopia

PONE-D-21-15858R1

Dear Dr. Gilano,

We’re pleased to inform you that your manuscript has been judged scientifically suitable for publication and will be formally accepted for publication once it meets all outstanding technical requirements.

Kind regards,

Mahfuzar Rahman, MD, PhD

Academic Editor

PLOS ONE
---

## [Editor Report · Acceptance letter]

21 Dec 2021

PONE-D-21-15858R1 

Geographical variation and associated factors of vitamin A supplementation among 6–59-month children in Ethiopia 

Dear Dr. Gilano:

I'm pleased to inform you that your manuscript has been deemed suitable for publication in PLOS ONE. Congratulations! Your manuscript is now with our production department. 

Kind regards, 

on behalf of

Dr. Mahfuzar Rahman 

Academic Editor

PLOS ONE